# Prevalence and determinants of low testosterone levels in men with type 2 diabetes mellitus; a case-control study in a district hospital in Ghana

**Dorcas Serwaa**[1,2]*, **Folasade Adenike Bello**[3,4], **Kayode O. Osungbade**[5], **Charles Nkansah**[6], **Felix Osei-Boakye**[7], **Samuel Kwasi Appiah**[6], **Maxwell Hubert Antwi**[8], **Mark Danquah**[7], **Tonnies Abeku Buckman**[9], **Ernest Owusu**[10]

1 Department of Obstetrics and Gynecology, College of Medicine, Pan African University, Yaoundé, Cameroun, 2 Institute of Life and Earth Sciences, University of Ibadan, Ibadan, Nigeria, 3 Department of Obstetrics and Gynaecology, University College Hospital, Ibadan, Oyo State, Nigeria, 4 Department of Obstetrics and Gynaecology, University of Ibadan, Ibadan, Oyo State, Nigeria, 5 Department of Community Medicine, College of Medicine, University College Hospital, Oyo State, Nigeria, 6 Department of Biomedical Laboratory Sciences, School of Allied Health Sciences, University for Development Studies, Tamale, Ghana, 7 Department of Medical Laboratory Technology, Faculty of Applied Science and Technology, Sunyani Technical University, Sunyani, Ghana, 8 Department of Medical Laboratory, Nkenkaasu District Hospital, Nkenkaasu, Ghana, 9 Department of Molecular Medicine, School of Medicine and Dentistry, Kwame Nkrumah University of Science and Technology, Kumasi, Ghana, 10 Department of Nursing and Midwifery, Methodist Health Training Institute, Afosu, Eastern Region, Ghana

* dserwaa0327@stu.ui.edu.ng

**Data Availability Statement:** All data are in the manuscript and/or Supporting information files.

## Abstract

Diabetes mellitus, an endocrine disorder, has been implicated in many including hypogonadism in men. Given the fact that diabetes mellitus is becoming a fast-growing epidemic and the morbidity associated with it is more disabling than the disease itself. This study sought to assess the prevalence of low testosterone levels and predictors in type 2 diabetes mellitus patients and non-diabetic men in a district hospital in Ghana. This hospital-based case-control study comprised 150 type 2 diabetics and 150 healthy men. A pre-structured questionnaire and patient case notes were used to document relevant demographic and clinical information. Venous blood sample of about 6 ml was taken to measure FBS, HbA1c, FSH, LH, and testosterone levels. All data were analyzed using STATA version 12 (STATA Corporation, Texas, USA). The overall hypogonadism in the study population was 48% (144/300). The prevalence of hypogonadism in type 2 diabetic subjects was almost three times more than in healthy men (70.7% vs 25.3%). The odds of having hypogonadism was lower in the men with normal weight and overweight with their underweight counterparts (AOR = 0.33, 95% CI; 0.12–0.96, p = 0.042) and (AOR = 0.29, 95% CI; 0.10–0.84, p = 0.023) respectively. Also, the odds of suffering from hypogonadism was lower in non-smokers compared with smokers (AOR: 0.16, 95% CI; 0.05–0.58, p = 0.005). Participants who were engaged in light (AOR: 0.29, 95% CI; 0.14–0.61, p = 0.001), moderate (AOR: 0.26, 95% CI; 0.13–0.54, p<0.001) and heavy (AOR: 0.25, 95% CI; 0.10–0.67, p = 0.006) leisure time activities had lower odds hypogonadal compared to those engaged in sedentary living. Type 2 diabetic men have high incidence of hypogonadism, irrespective of their

**Funding:** The author(s) received no specific funding for this work.

**Competing interests:** All authors declare that there is no competing interest.

baseline clinical, lifestyle or demographic characteristics. Smoking and sedentary lifestyle and BMI were associated with hypogonadism in the study population. Routine testosterone assessment and replacement therapy for high risk patients is recommended to prevent the detrimental effect of hypogonadism in diabetic men.

## Introduction

The relationship between sex hormones and Type 2 diabetes mellitus (T2DM) is of great concern in the health sector, given the fact that diabetes mellitus is becoming a fast-growing epidemic and the morbidity associated with it is more disabling than the disease itself. Millions of people around the world are diagnosed with T2DM, many more remain undiagnosed [1]. The world prevalence of diabetes mellitus among adults (aged 20–79 years) in 2010 was 6.4% and expected to increase to 7.7%, by 2030 [2]. It is estimated that in 2000, about 7, 146, 000 people in sub-Saharan Africa had diabetes mellitus, with a projected increase to 18, 645, 000 in 2030 [3, 4]. The prevalence of diabetes mellitus in Ghana is 6.4% and about 69.9% remain undiagnosed [4]. Diabetes mellitus has been implicated in male sexual dysfunction, libido dissociations, retrograde ejaculation, erectile dysfunction and lack of efficient endocrine control of spermatogenesis [5].

Hypogonadism is characterized by low serum testosterone concentration, followed by numerous clinical features like erectile dysfunction (ED), poor morning erection, low libido, loss of memory, physical decline in strength and health, difficulty in concentration and depression [6–8]. A number of studies have shown high incidence (30–80%) of hypogonadism in men with diabetes mellitus [9–11], a clear evidence of the association between type 2 diabetes and low serum testosterone. Although mediated by a variety of mechanisms, hypogonadism is more common in diabetic than in non-diabetic men in the Western world and in Asia compared to Africa. This could be attributable to the paucity of data on this issue in sub-Saharan African men.

There are very little documented data in Ghana on the prevalence of hypogonadism in both Type 2 diabetic patients and healthy men, because of limited resources and cost-ineffectiveness of screening all men. To the best of our knowledge only one study is available on hypogonadism in diabetes mellitus men in Ghana [5]. This study, therefore, determined the prevalence of hypogonadism among Type 2 diabetic men in a district hospital in Ghana, and non-diabetic controls.

## Subjects and methods

### Study design and study setting

This hospital based case-control study was conducted at Nkenkaasu District Hospital located in the Offinso-North district, in the Ashanti Region, Ghana. The total land area of the Offinso-North district is about 945.9 square kilometres and lies between latitude: 7˚20N. 6˚50S" and longitude: 1˚60W", 1˚45E". The majority of the inhabitants of this district are farmers [12]. The Nkenkaasu District Hospital serves as the main referral facility in the district and its neighboring villages. This Hospital records about 472 cases of diabetes annually, with 427 of them being T2DM, (per the outpatient department's report).

## Study participants

This study involved 150 type 2 diabetic men who had registered and receiving treatment at the diabetic clinic of Nkenkaasu Government Hospital and 150 control group comprised of apparently healthy blood donors and those visiting their relatives on admission. This study excluded patients on androgens, steroids medications, patients with a history of chronic renal failure, prostate cancer, prostatectomy and castrated men. The exclusion criteria for the healthy group was based on measurement of baseline fasting blood glucose (FBG) $\geq$ 7.0 mmol/l and HbA1c value $\geq$6% [13].

## Sample size determination

The necessary sample size was obtained by employing the Kelsey's for-

mula: $N_{cases-Kelsey} = \left[\frac{r+1}{r}\right] \frac{P(1-P)\left(Z_{\frac{\alpha}{2}}+Z_{\beta}\right)^2}{(p1-p2)^2}$, and $P = \left[\frac{p1+(r \, X \, p2)}{r+1}\right]$, where r is the ratio of T2DM to healthy controls, which is 1:1 in this study, $Z_{\frac{\alpha}{2}}$ represents the critical value of the normal dispersion at $\alpha/2$ (for this study at confidence level of 95%, $\alpha$ is 0.05 and the critical value is 1.96), $Z_{\beta}$ represents the critical value of the normal distribution at $\beta$ (this study used a power of 80%, $\beta$ is 0.2 and the critical value is 0.84. p1 represents the percentage of hypogonadism in Ghanaian men with diabetes, which is 35.2%, p2 is the percentage of hypogonadism in the control group, which is 6.7% according to Asare-Anane et al. [5], and p1-p2 is the smallest difference in proportions that is clinically important.

From the formula above, the minimum number of T2DM required for this study was 33 with corresponding controls of 33. However, this study employed 300 subjects: 150 T2DM patients and 150 healthy controls.

## Collection of information and patients selection

A structured questionnaire and patients' medical records were used to document relevant demographic and clinical history from the participants. Type 2 diabetes Mellitus was diagnosed through laboratory assessment based on current WHO diagnostic criteria (FBG $\geq$7.0 mmol/l or 126mg/dl) and HbA1c of <6.0% [13] and confirmed through a physician's recommendations.

## Anthropometric variables measurements

**Body mass index.**   Height to the nearest centimetre without shoes and weight to the nearest 0.1 kg in light clothing was estimated. Participants were weighed on a bathroom scale and their heights were measured with a wall-mounted ruler. Body mass index (BMI) was calculated by dividing weight (kg) by height squared (m$^2$). BMI was categorized as: <18.5 (underweight); 18.5 to 24.9 (normal weight); 25 to 29.9 (overweight); and $\geq$30 (obese) [13].

**Blood pressure (using Korotkoff 1 and 5).**   Blood pressure was measured by trained personnel using a mercury sphygmomanometer and a stethoscope. Measurements were taken from the left upper arm after participants sat >5 min in accordance with the recommendations of the American Heart Association. Duplicate measurements were taken with a 5-minutes rest interval between measurements and the mean value was recorded in mmHg. Hypertension was graded as normal when the systolic blood pressure (SBP) was >120 mm Hg and diastolic blood pressure (DBP) >80 mm Hg [14].

**Physical activities.**   Leisure-time physical activity was measured based on alternatives to the question "How physically active are you during your leisure time?". Subjects were characterized as having; sedentary leisure time if they perform the following activities reading,

watching television, stamp collecting or other sedentary activity; Light leisure-time physical activity if subjects engage in some walking, cycling, or other physical activity under at least four hours per week; Moderate leisure-time physical activity if they perform any of the following activities running, Swimming, tennis, aerobic, heavier gardening, or similar physical activity during at least 2 hours a week; and lastly, Heavy leisure-time physical activity if engaged in heavy training or competitions in running, skiing, swimming, football, etc, which is performed regularly and several times a week [15].

**Sexual health inventory for men (SHIM) questionnaire.** The SHIM questionnaire is a basic 5-point questionnaire on erectile dysfunction. Each answer is graded from 0 (no sexual activity or attempts at intercourse) to 5 (very good sexual function). The maximum score patients could obtain will be 25, the minimum was 1. Based on the SHIM questionnaire patients were divided into groups: ≤22; >22 = No ED. This questionnaire was completed by all patients [16].

**Health-related quality of life questionnaire.** A basic health-related quality of life questionnaire (EuroQoL group / EQ-5D questionnaire) was completed by all subjects. Although this questionnaire was chosen for its brevity and simplicity. Illiterate subjects were assisted in filling the questionnaire [17].

## Fasting blood glucose (FBG) measurement

Samples for FBG were analyzed using Accu-Chek Advantage Blood Glucose Monitoring System (AC; [3]Roche Diagnostics, Indianapolis, IN). Calibration of the instrument was performed at 7:00 am using the test kit glucose control solution. A fingertip capillary whole blood sample was collected from each subject after overnight fasting between 7:00 am and 10:00 am for determination of fasting blood sugar and diabetes mellitus [14].

## Blood sample collection

Whole venous blood of about 6 ml was obtained from each subject via a sterile venepuncture after overnight fasting between 7:00 am and 10: 00 am; 2.0 mL into EDTA for HbA1c and 4.0 mL dispensed into plain tubes at room temperature for 1 hr. after which the supernatant clear fluids were pipetted out to another tube. Plasma was separated after centrifugation (CENTRIFUGE 80–1, Japan). The clear serums were then pipetted into a clean dry test tube and separated into aliquots and frozen at -40 ˚c̣ until analyzed for LH, FSH and testosterone [1]. Hormonal estimation was determined by an enzyme-linked immunosorbent assay (ELISA) technique using automated ELISA washer (BIO-RAD, PW40) and ELISA reader (Mindray, MR-96A) [13, 14].

## Laboratory assay (FSH, LH and testosterone estimation)

All laboratory investigations were done at the Methodist Hospital Laboratory, Wenchi, Bono Region, Ghana. FSH is synthesized and secreted by gonadotrophs of the anterior pituitary gland [18]. FSH was determined using AccuDiag™ FSH ELISA Kit (Omega Diagnostic, Scotland, UK), FSH minimum detection range for AccuDiag™ FSH ELISA Kit is 0-200mIU/mL, specificity is 95% and sensitivity is 2.0mIU/ml. The normal reference range of FSH for the laboratory is 1.3–7.4 mIU/ml.

LH is used as an aid in the screening or monitoring of determination of evaluating fertility issues, function of reproductive organs (ovaries or testicles) [18]. LH with Kit (AccuDiag™ LH ELISA Kit: Omega Diagnostic, Scotland, UK). The minimum detection range LH test kit is 0-200mIU/mL, specificity of 95% and 2.0mIU/ml sensitivity. Normal LH value for men according to Methodist hospital laboratory is 1.8–7.4 mIU/ml.

AccuDiag™ Testosterone ELISA Kit (Omega Diagnostic, Scotland, UK) was used for testosterone estimation. The minimum detection limits for testosterone is 0–18 ng/ml, 95% specificity and 0.05 ng/ml sensitivity and normal range of testosterone according to the laboratory is 8.0–12.0 nmol/l.

Collecting and analysing serial serum samples eliminates variability resulting from the episodic secretion of hormones, hence this study evaluated double samples of each participant. Therefore, patients with borderline values were probably transiently suppressed by acute conditions or stress were captured appropriately upon repeat testing.

On the basis of normal ranges and international recommendations, hypogonadism in this study was described as total testosterone levels < 8 nmol/l, with or without signs and symptoms and total testosterone levels > 12 nmol/l was defined as eugonadal.

## Ethical consideration and informed consent

This study was conducted based on the Helsinki Declaration and study protocol, consent forms and participant information material were reviewed and approved by University of Ibadan/University Collage Hospital Ethics Committee (UI/EC/18/0621). Also, approval was obtained from the Offinso North District Assembly and the Nkenkaasu Hospital's Research and Ethics committee.

Written informed consent of individual participants was sought after the aims and objectives of the study had been thoroughly explained to them. Participants either signed or thumbprinted to give their consent, before the commencement of the study and they assured of the confidentiality of their data.

## Statistical analysis

The data were analysed using STATA version 12 (STATA Corporation, Texas, USA). Test for normality was performed using box plot and Kolmogorov-Smirnoff test. Frequencies, percentages, means, and standard deviations were calculated to enable comparison of characteristics between T2DM subjects and the healthy group. Besides the descriptive analysis, the independent t-test was used for the comparison between the categorical and continuous variables among the groups, results were expressed as mean ± SD. Correlation analysis was performed to estimate the relationships between testosterone levels and demographic and clinical variables. Binary logistic regression analysis was done and all independent variables at $p < 0.10$ were taken to multivariable logistic regression analysis by backward elimination to identify sociodemographic and lifestyle predictors of hypogonadism. The statistical significance of variables at final model was declared at $p < 0.05$ and 95% confidence level for the adjusted odds ratio.

## Results

Most of the study participants were aged 61–70 years 137(45.0), had normal weight 170(56.7), were engaged in farming 135(45.0), non-alcohol consumers 286(95.3), non-smokers 281 (93.7), had good Health-related Quality of Life (HRQoL), 256(85.3), engaged in moderate leisure time activity 103(34.3) and had erectile dysfunction 257(85.7). There was statistically significant difference between the type 2 diabetic men and the healthy controls when categorized by age (p = 0.003), body mass index (p<0.001), smoking (p<0.001), HRQoL (p = 0.014) leisure time activity (p<0.001) and erectile function status (p<0.001) (Table 1).

There was no statistically significant difference between the ages the type 2 diabetic men and the healthy controls (58.25±9.71 vs 56.34±9.40, p = 0.084). The body mass index (BMI) of the healthy controls was significantly higher than the type 2 diabetic men (24.01±3.38 and

**Table 1. Baseline characteristics of the study participants.**

| Characteristics | Total | Type 2 Diabetics | Healthy controls | P value |
|---|---|---|---|---|
| | N (%) | N (%) | N(%) | |
| **Age (years)** | | | | |
| 41–50 | 88(29.3) | 35(23.3) | 53(35.3) | 0.003* |
| 51–60 | 77(25.7) | 33(22.0) | 44(29.3) | |
| 61–70 | 137(45.0) | 82(54.7) | 53(35.3) | |
| **BMI ($kg/m^2$)** | | | | |
| Underweight | 25(8.3) | 20(13.3) | 5(3.3) | <0.001* |
| Normal weight | 170(56.7) | 89(59.3) | 81(54.0) | |
| Overweight | 91(30.3) | 32(21.3) | 59(39.3) | |
| Obese | 14(4.7) | 9(5.0) | 5(3.3) | |
| **Occupation** | | | | |
| Farming | 135(45.0) | 60(40.0) | 75(50.0) | 0.107 |
| Trading | 47(15.7) | 37(24.7) | 21(140.) | |
| Civil Servants | 58(19.3) | 23(15.3) | 24(16.0) | |
| Unemployed | 60(20.0) | 30(20.0) | 30(20.0) | |
| **Alcohol consumption** | | | | |
| Yes | 14(4.7) | 9(6.0) | 5(3.3) | 0.206 |
| No | 286(95.3) | 141(94.0) | 145(96.7) | |
| **Smoking** | | | | |
| Yes | 19(6.3) | 19(12.7) | 0(0.0) | <0.001* |
| No | 281(93.7) | 131(87.3) | 150(100.0) | |
| **HRQoL** | | | | |
| Poor | 44(14.7) | 30(20.0) | 14(9.3) | 0.014* |
| Good | 256(85.3) | 120(80.0) | 136(90.7) | |
| **Leisure Time Activity** | | | | |
| Sedentary | 75(25.0) | 56(37.3) | 19(12.7) | <0.001* |
| Light | 92(30.7) | 36(24.0) | 56(37.3) | |
| Moderate | 103(34.3) | 44(29.3) | 59(37.3) | |
| Heavy | 30(10.0) | 14(9.3) | 16(10.7) | |
| **Erectile function status** | | | | |
| No ED | 43(14.3) | 4(2.7) | 39(26.0) | <0.001* |
| ED | 257(85.7) | 146(97.3) | 111(74.0) | |

Body Mass Index (BMI), Health-related Quality of Life (HRQoL), Erectile Dysfunction (ED).

23.05±4.04, p = 0.026). The type 2 diabetic men had significantly higher systolic blood pressure (SBP) (145.76±24.77 vs 134.94±25.36, p<0.001), diastolic blood pressure (DBP) (86.74±12.91 vs 82.15±9.01, p<0.001), fasting blood sugar (FBS) (10.33±5.57 vs 5.84±0.61, p<0.001) and glycated hemoglobin (HbA1c) levels (8.06±2.58 vs 4.78±0.59, p<0.001) compared with the healthy controls. The biochemical analysis revealed, mean serum follicle stimulating hormone (FSH) (8.85±5.05 vs 7.19±4.68, p = 0.003), luteinizing hormone (LH) (7.08±3.90 vs 6.18±3.58, p = 0.017) and testosterone (13.01±7.85 vs 7.66±5.45, p<0.001) levels were significantly higher in the healthy controls relative to the type 2 diabetic men (Table 2).

**Table 2. Clinical and hormonal parameters of the study population.**

| Parameters | Type 2 Diabetics (n = 150) | Healthy controls (n = 150) | 95% CI | P value |
|---|---|---|---|---|
| Age (yrs) | 58.25±9.71 | 56.34±9.40 | -0.258–4.084 | 0.084 |
| BMI (Kg/m$^2$) | 23.05±4.04 | 24.01±3.38 | -1.810-(-0.118) | 0.026* |
| SBP (mmHg) | 145.76±24.77 | 134.94±25.36 | 5.124–16.516 | <0.001* |
| DBP (mmHg) | 86.74±12.91 | 82.15±9.01 | 2.047–7.126 | <0.001* |
| FBS (mmol/L) | 10.33±5.57 | 5.84±0.61 | 3.588–5.394 | <0.001* |
| HbA1C (%) | 8.06±2.58 | 4.78±0.59 | 2.854–3.707 | <0.001* |
| FSH (mIu/ml) | 7.19±4.68 | 8.85±5.05 | -2.767-(-0.555) | 0.003* |
| LH (mIu/ml) | 6.18±3.58 | 7.08±3.90 | -1.642-(-0.162) | 0.017* |
| Testosterone (nmol/l) | 7.66±5.45 | 13.01±7.85 | -6.887-(-3.818) | <0.001* |

(Mean± SD): Body Mass Index (BMI), Diastolic Blood Pressure (DBP), Systolic Blood Pressure (SBP), Fasting Blood Sugar (FBS), Follicle Stimulating Hormones (FSH), Luteinizing Hormone (LH) and * mean difference is significant (p<0.05).

## Correlation of testosterone (T) with fasting blood sugar among the participants

Fig 1 shows the correlation between testosterone levels and fasting blood sugar (FBG) levels among the study participants. A statistically significant positive correlation was observed between free testosterone levels and FBS (r = 0.233, p <0.001).

## Correlation of testosterone (T) with glycated hemoglobin (HbA1c) among the participants

Fig 2 shows the correlation between testosterone levels and glycated haemoglobin (HbA1c) levels among the study participants. A statistically significant inverse correlation

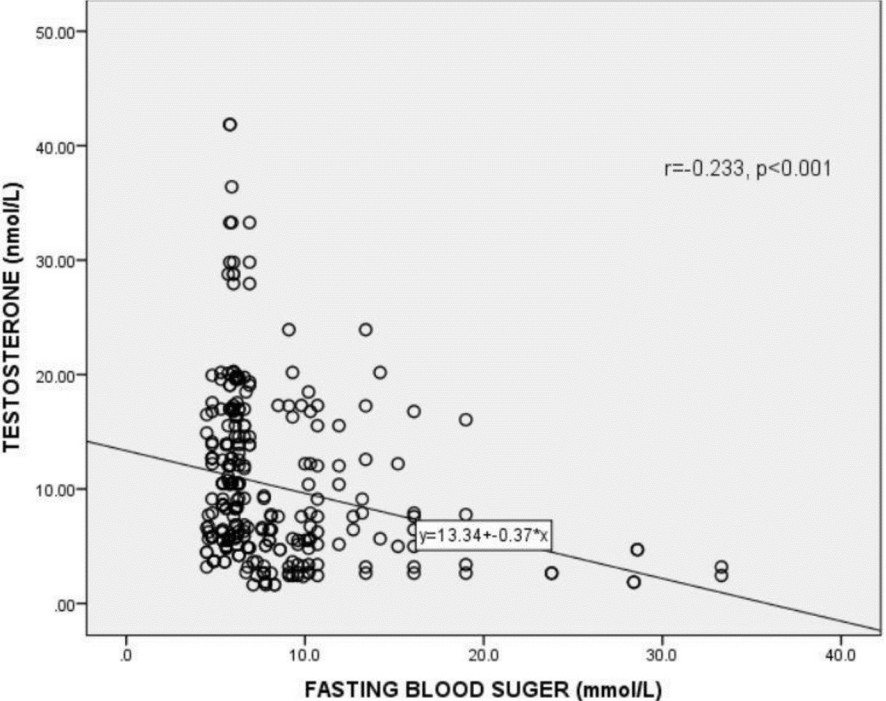

**Fig 1. Correlation between testosterone levels and fasting blood glucose.** r = Correlation coefficient. *p*<0.05 was considered statistically significant.

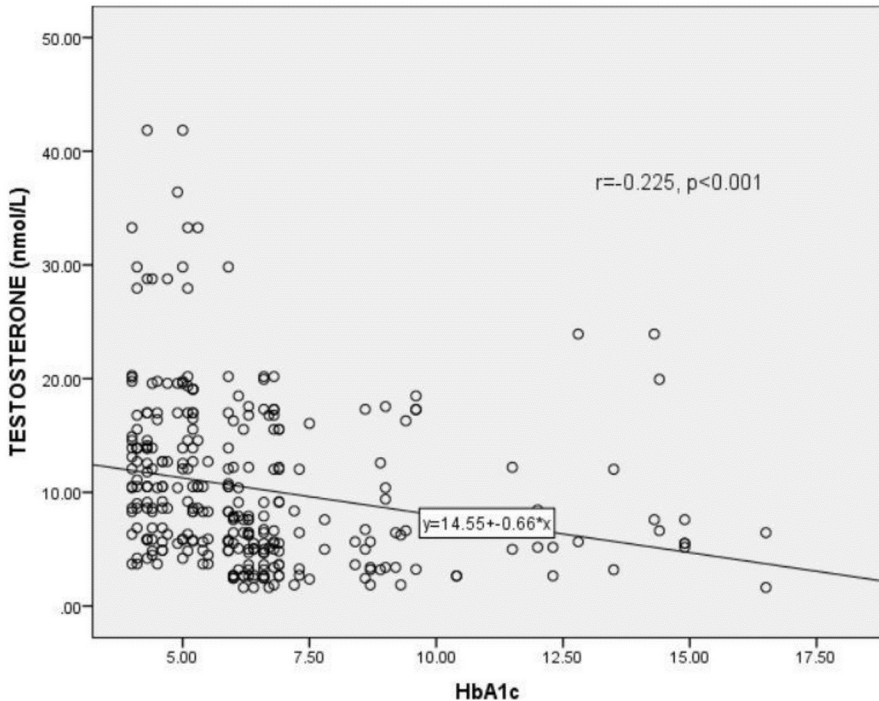

**Fig 2. Correlation between testosterone levels and glycated hemoglobin (HbA1c) levels among the study participants.** HbA1c = Glycated haemoglobin, r = Correlation coefficient. $p < 0.05$ was considered statistically significant.

existed between the testosterone levels and HbA1c levels of the study participants (r = 0.225, p <0.001).

## Prevalence of hypogonadism in the study population

Fig 3 shows the percentage distribution of testosterone levels of the diabetic and non-diabetic subjects. The overall hypogonadism in the study population was 48% (144/300). The prevalence of hypogonadism in the type 2 diabetic subjects (T< 8 nmol/l) was almost three times more than healthy men (70.7% vs 25.3%). Also, 9 (6.0%) and 39 (26%) had testosterone levels between 8–12 nmol/L for the type 2 diabetic men and non-diabetic men respectively. In addition, 35 (23.3%) and 73 (48.7%) were eugonadal for type 2 diabetic and non-diabetic men respectively. Chi square analysis revealed a statistically significant positive association between Type 2 Diabetes Mellitus and hypogonadism as indicated by the p-value from the chi-square analysis (p<0.001).

The distribution of categorized testosterone levels for the diabetic men was 70.7%, 6.0% and 23.3% for <8 nmol/L, between (8–12 nmol/L and >12 nmol/L respectively. Distribution of categorized testosterone levels for the non-diabetic group was 24.5%, 26.0% and 48.7% for < 8 nmol/L, between (8–12 nmol/L and >12 nmol/L.

## Clinical and hormonal parameters of type 2 diabetic and the healthy hypogonadal men

The mean age between the type 2 diabetic and the apparently healthy hypogonadal men were not significantly different (57.48±9.62 vs 56.92±9.94, p = 0.761). Similarly, no significant

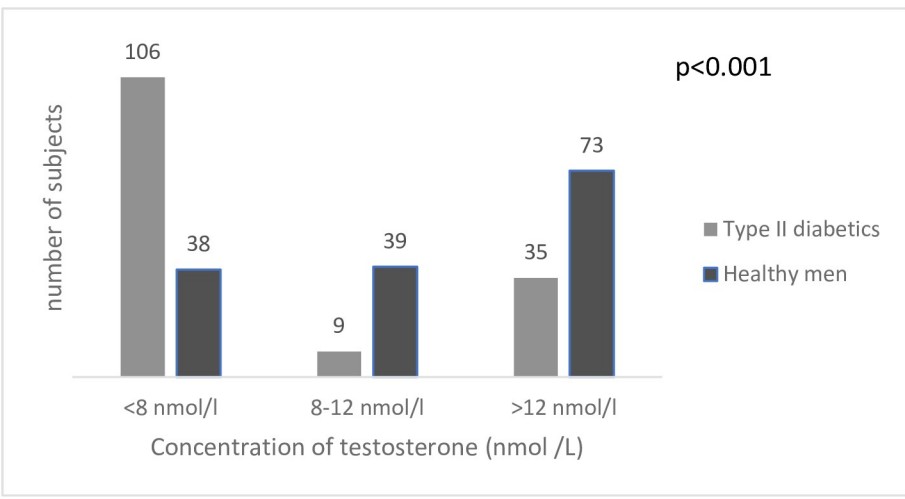

**Fig 3. Percentage distribution of categorized testosterone levels.**

differences were observed between BMI (22.94±3.98 vs 24.18±3.92, p = 0.099), SBP (144.42 ±23.39 vs 134.50±30.81, p = 0.076) and DBP (85.57±12.76 vs 83.32±7.30, p = 0.192) of type 2 diabetic and the apparently healthy hypogonadal men. With respect to the LH levels, no significant difference was observed between type 2 diabetic and the apparently healthy hypogonadal men (5.80±3.90 vs 5.80±4.65, p = 0.996). The healthy hypogonadal men were significantly different from the type 2 diabetic hypogonadal men with reference to FBG (10.75±6.21 vs 5.82 ±0.81, p<0.001), HbA1c (8.06±2.68 vs 4. 87±0.61, p = <0.001) and FSH (7.61±5.16 vs 10.62 ±5.33, p = 0.003) level (Table 3).

## Binary and multivariable logistic regression about determinants of hypogonadism among the study participants

In both the bivariate and multivariate logistic regression analyses, BMI, smoking and leisure time activity (sedentary lifestyle) were associated with hypogonadism in the study population. The odds of having hypogonadism was lower in the men with normal weight and overweight with their underweight counterparts (AOR = 0.33, 95% CI; 0.12–0.96, p = 0.042) and (AOR = 0.29, 95% CI; 0.10–0.84, p = 0.023) respectively. Also, the odds of suffering from

**Table 3. Clinical and hormonal parameters of type 2 diabetic and healthy hypogonadal men.**

| Parameters | Type 2 Diabetics | Healthy controls | 95% CI | P value |
|---|---|---|---|---|
| Age (yrs) | 57.48±9.62 | 56.92±9.94 | -3.067–4.188 | 0.761 |
| BMI (Kg/m$^2$) | 22.94±3.98 | 24.18±3.92 | -2.732–0.242 | 0.099 |
| SBP (mmHg) | 144.42±23.39 | 134.50±30.81 | -1.088–20.937 | 0.076 |
| DBP (mmHg) | 85.57±12.76 | 83.32±7.30 | -10146-5.646 | 0.192 |
| FBG (mmol/L) | 10.75±6.21 | 5.82±0.81 | 3.707–6.153 | <0.001* |
| HbA1C (%) | 8.06±2.68 | 4. 87±0.61 | 2.636–3.738 | <0.001* |
| FSH (mIu/ml) | 7.61±5.16 | 10.62±5.33 | -4.950-(-1.060) | 0.003* |
| LH (mIu/ml) | 6.31±3.95 | 7.09±2.82 | -1.968–0.401 | 0.192 |

(Mean± SD): Body Mass Index (BMI), Diastolic Blood Pressure (DBP), Systolic Blood Pressure (SBP), Fasting Blood Sugar (FBS), Follicle Stimulating Hormones (FSH), Luteinizing Hormone (LH) and * mean difference is significant (p<0.05).

**Table 4. Binary and multivariate logistic regression of sociodemographic, clinical and lifestyle factors associated of hypogonadism in the study participants.**

| Characteristics | HYPOGONADISM | | Unadjusted OR[95% CI] | P value | Adjusted OR[95% CI] | P value |
|---|---|---|---|---|---|---|
| | Yes | No | | | | |
| | N = 144 | N = 156 | | | | |
| **Age (years)** | | | | | | |
| 41–50 | 38(43.2) | 50(56.8) | 1 | | 1 | |
| 51–60 | 42(54.5) | 35(45.5) | 1.58[0.85–2.92] | 0.146 | 1.80[0.90–3.90] | 0.094 |
| 61–70 | 64(47.4) | 71(52.6) | 1.89[0.69–2.04] | 0.536 | 1.20[0.57,2.40] | 0.666 |
| **BMI ($kg/m^2$)** | | | | | | |
| Underweight | 17(68.0) | 8(32.0) | 1 | | 1 | |
| Normal weight | 84(49.4) | 86(50.6) | 0.46[0.19–1.12] | 0.088 | 0.33[0.12–0.96] | 0.042* |
| Overweight | 36(39.6) | 55(60.4) | 0.31[0.12–1.79] | 0.014* | 0.29[0.10–0.84] | 0.023* |
| Obese | 7(50.0) | 7(50.0) | 0.47[0.12–1.80] | 0.271 | 0.63[0.12,3.21] | 0.575 |
| **Occupation** | | | | | | |
| Unemployed | 24(40.0) | 36(60.0) | 1 | | 1 | |
| Farming | 70(51.9) | 65(48.1) | 1.62[0.87–2.99] | 0.128 | 2.11[0.90–4.93] | 0.086 |
| Trading | 33(56.9) | 25(43.1) | 1.98[0.95–4.12] | 0.068 | 1.93[0.75–4.98] | 0.176 |
| Civil Servants | 17(36.2) | 30(63.8) | 0.85[0.39–1.87] | 0.686 | 0.82[0.30–2.21] | 0.688 |
| **Alcohol consumption** | | | | | | |
| Yes | 8(57.1) | 6(42.9) | 1 | | 1 | |
| No | 136(47.6) | 150(52.4) | 0.68[0.43–2.01] | 0.485 | 0.62[0.18–2.16] | 0.458 |
| **Smoking** | | | | | | |
| Yes | 15(78.9) | 129(45.9) | 1 | | 1 | |
| No | 4(21.1) | 152(54.1) | 0.23[0.07,0.70] | 0.010* | 0.16[0.05–0.58] | 0.005* |
| **HRQoL** | | | | | | |
| Poor | 20(45.5) | 24(54.5) | 1 | | 1 | |
| Good | 124(48.4) | 132(51.6) | 1.10[0.59–2.14] | 0.715 | 1.30[0.63–2.68] | 0.342 |
| **Leisure Time Activity** | | | | | | |
| Sedentary | 49(65.3) | 26(34.7) | 1 | | 1 | |
| Light | 40(43.5) | 52(56.5) | 0.41[0.22–0.78] | 0.005* | 0.29[0.14–0.61] | 0.001* |
| Moderate | 43(41.7) | 60(58.3) | 0.38[0.21–0.70] | 0.002* | 0.26[0.13–0.54] | <0.001* |
| Heavy | 12(40.0) | 18(60.0) | 0.35[0.15–0.85] | 0.019* | 0.25[0.10–0.67] | 0.006* |
| **Erectile function status** | | | | | | |
| No ED | 15(34.9) | 28(65.1) | 1 | | 1 | |
| ED | 129(50.2) | 128(48.9) | 1.88[0.96–3.69] | 0.066 | 1.30[0.60–2.78] | 0.517 |

Body Mass Index (BMI), Health-related Quality of Life (HRQoL), Erectile Dysfunction (ED).

hypogonadism was lower in non-smokers compared with smokers (AOR: 0.16, 95% CI; 0.05–0.58, p = 0.005). Participants who were engaged in light (AOR: 0.29, 95% CI; 0.14–0.61, p = 0.001), moderate (AOR: 0.26, 95% CI; 0.13–0.54, p<0.001) and heavy (AOR: 0.25, 95% CI; 0.10–0.67, p = 0.006) leisure time activities had lower odds hypogonadal compared to those engaged in sedentary living (Table 4).

## Discussion

The testosterone hormone has a major impact on men's overall health and well-being. This hospital-based case-control study sort to ascertain the prevalence and determinants of low testosterone levels in Type 2 diabetes mellitus Ghanaian men compared to non-diabetic controls.

The findings revealed that, the mean age of diabetic men in this sample was not substantially different from that of non-diabetic men. The main occupation of the inhabitants of this district is farming, therefore it was not surprising that most of the study participants were farmers. Compared to the type 2 diabetic men, the healthy men recorded the highest number of smokers and alcohol consumers. It is likely that health education provided to diabetic patients during their routine clinics encouraged some to quit smoking and drinking excessively as part of lifestyle modification. More so, the use of questionnaires to assess smoking and alcohol consumption status may have a social desirability issue diminishing response rate. The majority of the diabetic men had a good quality of life compared with healthy controls. It is plausible that the diabetic patients are educated on the implication of worrying and overthinking on blood glucose control, hence the observed difference. A sedentary lifestyle was observed more in the healthy group compared with the diabetics. This is probably because of sedentary lifestyle modification in most diabetic men due to their condition.

Erectile dysfunction rate was quite high in our study subjects irrespective of their diabetic status and it seems normal at this age group. According to a recent analysis on the prevalence of sexual dysfunction the prevalence of ED was 1%–10% in men younger than 40 years, 2%–9% among men between 40 and 49 years, and it increased to 20%–40% among men between 60–69 years, reaching the highest rate in men older than 70 years (50%–100%) [19]. In the Massachusetts Male Aging Study, diabetic men showed a threefold probability of having ED than men without diabetes; moreover, the age-adjusted risk of ED was doubled in diabetic men compared with those without diabetes [20]. Similar erectile dysfunction rates were also found in France, where 39% of men aged 18 to 70 reported erectile dysfunction [21]. Another report by Thai Erectile Dysfunction Epidemiological Study Group (TEDES) among men aged 40 to 70 revealed an erectile dysfunction prevalence of 37.5% [22]. The findings of this study defining the age group for erectile dysfunction does not rule out ED at early or late stage, therefore categorizing our age group around 40–70 years could not change already known facts the most important factor is the stage of the diabetes. Among the diabetic patients, as age increases and/ or the condition progresses, the risk of developing peripheral neuropathy, hypertension, and impotency would also elevate, which might be the reason for an increased odds of ED.

The findings of the study depicted that the T2DM cohort had a lower BMI than the control cohort. Even though the average BMI of the diabetic men was significantly lower than the healthy men, both were within the normal range. This contradicts the results of [17] that reported that males with diabetes have a higher average BMI than their non-diabetic counterparts. In the pathogenesis of T2DM, lower body mass index (BMI) is consistently associated with reduced type 2 diabetes risk, among people with varied family history, genetic risk factors and weight, while in established T2DM patients weight loss has been shown to meliorate glycaemic control, with severe calorie restraint even reversing the progression of T2DM [23, 24]. The mechanisms for this BMI paradox are not fully understood, but proposed explanations include T2DM individuals lose weight and become frail as a result of underlying illnesses that cause wasting. A study by Peyrot et al. [25], into possible psychological barriers to diabetes care, also found that many participants with T2DM felt very anxious and ashamed about their weight and thus reducing their weight reduces their experience of weight stigma.

Another possible explanation is the genetic predisposition in T2DM patients. According to Habbu et al. [26], more South Asians developed T2DM at BMI below 30 kg/m$^2$ (38%) than White Europeans (26%) and African-Caribbeans (29%). This suggest a possible low BMI among T2DM patients in our study subjects. Lastly, life style interventions that target diets and weight-loss have shown demonstrable benefit for reducing the risk of type II diabetes in high-risk and pre-diabetic individuals but have not been well-studied in people at lower risk of diabetes. These findings suggest that all individuals can substantially reduce their type 2

diabetes risk through weight loss, and support the broad deployment of weight loss interventions to individuals at all levels of diabetes risk as a public health measure [24]. In our present study since our diabetic cohorts visit the clinic regularly and visit the dietician, there might be measures going on to help reduce the risk of the infection, which might not be seen in the control cohort who might be moving around freely without any restrictions on diet. There was a proportional significant difference between diabetic and non-diabetic subjects for normal weight and overweight. According to Hu et al. [27], overweight is the single most significant defining factor of type 2 diabetes; therefore, it was not shocking to see that more diabetic men were overweight and obese than healthy men.

The mean systolic and diastolic blood pressures were significantly higher in diabetic men. Previous studies have asserted that the most adverse outcome of type 2 diabetes mellitus is hypertension because of the complications like diabetic nephropathy, increased exchangeable sodium, insulin resistance and peripheral vascular resistance associated with the disease [28, 29]. Other previous studies have also shown a significant mean difference in body mass index, systolic and diastolic blood pressures and fasting blood glucose between diabetic men and control groups [30, 31]. Also, a significant elevation of FBS and HbA1c levels was identified in the diabetic group compared to the non-diabetics. The elevated FBS and poor glycaemic control have been found to be directly proportional to the severity of the diabetes mellitus. The increasing FBS level and poor glycaemic control in the diabetic group were also in agreement with many other research findings [1, 18, 32].

The biochemical findings of this study showed a highly significant reduction in FSH, LH and testosterone levels were observed in the diabetic group compared to the healthy group. This agrees with the finding of Dhindsa et al. [33], which stated a significantly lower FSH and LH concentrations in the diabetic group compared with the controls. This finding also agrees in part with a study conducted by Hussein & Al-qaisi [1], except that their study reported an increased LH level in diabetic group. The diminished gonadotropin secretion in our diabetic subjects might have resulted in insufficient testicular stimulation, hence a reduction in testosterone secretion. Low testosterone levels in the diabetic men may interfere with potency, spermatogenesis and consequently fertility. Our findings showed a significant inverse relationship between low testosterone and FBS and HbA1c. Glycaemia is known to affect Leydig cell function directly causing primary hypogonadism and therefore the association between FBS, HbA1c and reduced total testosterone concentration might be an adverse effect of glycaemia on the testicular microvasculature. The low testosterone levels observed could also be a result of glucose not reaching the cells due to insulin insensitivity, to provide enough energy for the various metabolic processes involved in maintaining testosterone levels [9].

The study depicted that, most of our study participants had low testosterone level. Most of the study participants were at the age group of 40 to 70 years. Studies have shown that mean testicular volume and gonadal function diminishes at this ages. The mean testicular volume tends to increase between 11 and 30 years of age, remains constant between 30 and 60 years of age, and decreases gradually every year after age 60 [34]. Few data on hypogonadism in aging men are available because of the deficiency of evidence regarding the exact criteria for distinguishing testosterone deficiency in older men who do not have pathological hypogonadism [35–37]. The Massachusetts Male Aging Study, using both total testosterone and calculated free testosterone, gave crude prevalence estimates for hypogonadism in men from age 40 to 69 years, ranging at baseline from 6.0 to 12.3% [38]. This is in accordance with the results of this study. Mahmoud et al. [39], found the mean testicular volume in men over 75 years to be 31% less than in men between 18 and 40 years of age. This difference is associated with significantly higher mean serum levels of gonadotropins and lower serum free testosterone. Interestingly, when the incidence of hypogonadism was determined by decades, nearly all of the categories

of illness were more prevalent in men aged 50–70 years [40], which is consistent with our study.

This study showed that hypogonadism is a common defect in type 2 diabetic men, affecting more than half of the study group, irrespective of their baseline clinical, lifestyle or demographic characteristics. Drugs commonly implicated to induce mild to moderate reduction in serum testosterone levels include B blocker antihypertensive and anticholesterols (statins) which are mostly prescribed for the management of hypertension in T2DM and hypertension co-morbidity [41]. The high prevalence of hypogonadism in the study population raises important issues about its possible consequences the sexual, reproductive and general health (libido, erectile dysfunction, body musculature, abdominal adiposity, bone density, mood, and cognition) of our study population. The Endocrine Society has recommended routine examination and replacement therapy for diabetic patients [42]. However, most facilities in Ghana have not adopted it because of limited resources and cost-ineffectiveness of screening all men for hypogonadism.

A study conducted in Ghana reported a hypogonadism prevalence of 35.2% in men with type 2 diabetes [5]. Unlike this previous study which was conducted in urban setting and a teaching hospital, our study was carried out in a peri-urban setting and district hospital. Other previous studies have shown a prevalence of 30–80% in men with type 2 diabetes [43–45]. The disparities in prevalence could be attributed to the difference in population examined, the definition used for the diagnosis of hypogonadism and the sample size.

Clinical and hormonal parameters of type 2 diabetic and healthy hypogonadal men were again determined. The mean age, BMI, SBP, DBP and LH between the type 2 diabetic and the apparently healthy hypogonadal men were not significantly different. This finding did not agree with a study that have linked decreasing testosterone levels with aging even in healthy men [11]. The multivariate analysis from our study indicated that BMI, smoking and leisure time activity (sedentary lifestyle) were associated with hypogonadism in the study population. Most cross-sectional studies have shown a positive association between smoking and total or free testosterone levels [46, 47]. Also, some studies have shown a significant association between BMI and hypogonadism [5, 48, 49], while another had reported no relationship between testosterone and BMI [50]. The lack of physical exercise activity contributes to lowering down the testosterone hormone level and indeed beside the effect of obesity [9].

There are few limitations in the study. Firstly, this was a case-control design, which made it impossible to determine whether diabetes preceded or followed the decline in hormone levels. The study did not measure Estradiol (E2) to ascertain any trend because it has been shown that low testosterone levels in diabetes could also be as a result of their increased conversion to E2. The study was also limited by the advanced age of the participants (41–70 years), hence the high prevalence of hypogonadism may have been masked by the age bracket of the study participants. To help address this limitation, we recommend future studies to consider participants below 40 years of age. In the face of these limitations, this study gives significant data for the occurrence and predictors of hypogonadism among Ghanaian men with T2D in the district.

## Conclusion

The study recorded a worrying prevalence of hypogonadism even among the healthy control group but with Type 2 diabetic men having a high incidence of hypogonadism. This study also demonstrates that FSH and LH concentrations are certainly lower in a relatively large number of the males with type II diabetes compared with the healthy men. Body Mass Index, smoking and leisure time activity (sedentary lifestyle) were associated with hypogonadism in the study

population. Given a large number of individuals with diabetes worldwide, the high prevalence of hypogonadism in type 2 diabetes raises important issues about its possible consequences on the sexual, reproductive and general health of our study population. A further study is recommended to carry out to holistically assess the prevalence of gonadal deficiencies in subjects with prediabetes patients aged between 20–40 years at the district hospital.

## Supporting information

**S1 Data. All data and related metadata underlying the findings reported are provided as part of the submitted article.**
(RAR)

**S1 Checklist. Checklist of items that should be included in reports of *case-control studies*.**
*Give information separately for cases and controls in case-control studies, NA = Not Applicable.
(DOCX)

## Acknowledgments

We are grateful for the immense contributions of the staff of Nkenkaasu District Hospital and Wenchi Methodist Hospital Laboratories, not forgetting our participants.

## Author Contributions

**Conceptualization:** Dorcas Serwaa.

**Data curation:** Dorcas Serwaa, Folasade Adenike Bello, Kayode O. Osungbade, Charles Nkansah, Samuel Kwasi Appiah, Maxwell Hubert Antwi.

**Formal analysis:** Dorcas Serwaa, Kayode O. Osungbade, Charles Nkansah, Felix Osei-Boakye, Samuel Kwasi Appiah, Maxwell Hubert Antwi, Mark Danquah, Tonnies Abeku Buckman.

**Investigation:** Dorcas Serwaa, Folasade Adenike Bello, Kayode O. Osungbade, Mark Danquah.

**Methodology:** Dorcas Serwaa, Folasade Adenike Bello, Charles Nkansah, Felix Osei-Boakye, Samuel Kwasi Appiah, Maxwell Hubert Antwi.

**Supervision:** Dorcas Serwaa, Folasade Adenike Bello, Kayode O. Osungbade.

**Validation:** Dorcas Serwaa.

**Writing – original draft:** Dorcas Serwaa, Charles Nkansah, Felix Osei-Boakye, Maxwell Hubert Antwi, Mark Danquah, Tonnies Abeku Buckman, Ernest Owusu.

**Writing – review & editing:** Dorcas Serwaa, Folasade Adenike Bello, Kayode O. Osungbade, Charles Nkansah, Felix Osei-Boakye, Maxwell Hubert Antwi, Mark Danquah, Tonnies Abeku Buckman, Ernest Owusu.

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
