## [Decision Letter · Decision Letter 0]

6 Aug 2021

PGPH-D-21-00300

Prevalence and determinants of low testosterone levels in Type II diabetes mellitus men; a case-control study in a district hospital in Ghana

Dear Dr. Serwaa,

Thank you for submitting your manuscript to PLOS Global Public Health. After careful consideration, we feel that it has merit but does not fully meet PLOS Global Public Health’s publication criteria as it currently stands. Therefore, we invite you to submit a revised version of the manuscript that addresses the points raised during the review process.

Please consider incorporating the comments of all reviewers.

We look forward to receiving your revised manuscript.

Kind regards,

Palash Chandra Banik, MPhil

Academic Editor

Journal Requirements:

Additional Editor Comments (if provided):

- Please ensure the language editing throughout the manuscript.

Reviewers' comments:

Reviewer's Responses to Questions

**Comments to the Author**

1. Does this manuscript meet PLOS Global Public Health’s publication criteria? Is the manuscript technically sound, and do the data support the conclusions? The manuscript must describe methodologically and ethically rigorous research with conclusions that are appropriately drawn based on the data presented.

Reviewer #1: Yes

Reviewer #2: Yes

Reviewer #3: Partly

2. Has the statistical analysis been performed appropriately and rigorously?

Reviewer #1: Yes

Reviewer #2: Yes

Reviewer #3: Yes

3. Have the authors made all data underlying the findings in their manuscript fully available (please refer to the Data Availability Statement at the start of the manuscript PDF file)?

Reviewer #1: Yes

Reviewer #2: Yes

Reviewer #3: Yes

4. Is the manuscript presented in an intelligible fashion and written in standard English?

Reviewer #1: No

Reviewer #2: Yes

Reviewer #3: No

5. Review Comments to the Author

Reviewer #1: Authors try to find the prevalence and detrimental effect of testosterone levels in type 2 diabetes patients by case control study population. Previously, many studies were reported the increased testosterone levels in type 2 diabetes patients like "Testosterone level and risk of type 2 diabetes in men: a systematic review and meta-analysis"

however, the present study findings are specific to the ghana city. The provided information is not a new information and novelty of the study was not observed.

Few minor corrections:

Type II diabetes need to be replaced with type 2 diabetes mellites.

articles needs to be written well Like, line 25 The should be capital

Line 374 it is written that cross sectional study and before it is written as case control study. Need a clarity on this.

Table 3 Odd ratio is missing.

Reviewer #2: Dr. Serwaa and colleagues present a detailed case control study of 300 men with type 2 diabetes mellitus and average normal weight for low testosterone in a district hospital within a peri-urban setting, finding significant associations in this populations as shown in a prior urban setting in Ghana. Their results contribute to the literature associating type 2 diabetes and dysglycemia with male hypogonadism, in this case in a peri-urban setting of Nkenkaasu Hospital in Nkenkaasu, Ghana, West Africa in a distinct population from prior studies that contributes to the global understanding of male hypogonadism prevalence. A particular finding of interest is that both their case and control cohorts of moderate size where of normal weight on average, with their diabetes group having suboptimal glycemic control by A1c (8.06) and lower BMI (23.05) than their control group. Strengths also include measurement of pituitary hormones LH and FSH which are lower in the diabetes group, suggesting a relative secondary hypogonadism compared to the control cohort and consistent with hypogonadism occurring in type 2 diabetes. The results presented are complete, the statistical analysis is rigorous and the study was conducted following Declaration of Helsinki ethical principles. Figures 1 and 2 are particularly informative as the convey the variation in testosterone levels in relation to fasting blood glucose and HbA1c. The discussion is nicely written and cites relevant literature including articles documenting similar associations in the Ghanian population as well as the Endocrine Society guidelines as applied in this population. The authors acknowledge limitations of the study including difficulty measuring testosterone in this community due to lack of treatment access, and lack of estradiol measurement.

--Major Issues

*Title: The current title is worded awkwardly and should use person-first language such as “"Prevalence and determinants of low testosterone levels in men with Type II diabetes mellitus; a case-control study in a district hospital in Ghana"

*Assay: The methodological description of the assay AccuDiag Testosterone Assay Kit suggests using a total testosterone measurement, but the authors later refer to free testosterone in the results section. Could the authors clarify if they measured total, free or both testosterone levels? Male hypogonadism in insulin resistance partially involves a decrease in SHBG lowering total testosterone levels, but preserving intact free testosterone levels. This distinction is important as the ELISA assay is different than the commonly accepted LC-MS assays used in other parts of the world that report both free and total testosterone levels.

*Results – One of the more interesting findings of the study is that the diabetes cohort had a lower BMI than the control cohort. Could the authors comment on reasons as to why this might be a difference? Could it be attributed to stage of type 2 diabetes, poor control and subsequent glucosuria, or other factors?

*Results – Dysglycemia occurs in a spectrum, and the authors likely observed similar gonadal deficiencies in subjects with prediabetes as part of clinical care. Could the authors comment on how the observed association would be affected by prediabetes in their population, or as a future study?

*Reporting standards – As a reviewer I would ask the authors to complete the STROBE statement checklist for ease of review (https://www.strobe-statement.org/index.php?id=strobe-home). Most of the elements are included in the document, but the checklist would simplify review and compliance checking. Unfortunately I was not able to review the supplemental material in the RAR or data.sav file provided.

--Minor Issues

*I would suggest organizing the manuscript with the text first and tables at the end per usual manuscript submission guidelines.

*There are minor grammatical spelling errors that can be corrected with editorial assistance. The list below is not exhaustive:

Page 13, lines 216-217: Sentence should be corrected to “The body mass index (BMI) of the healthy controls was significantly higher than...”

Page 15, line 258: Sentence should start with “The distribution…"

Page 22: Line 326: Sentence should be corrected to “Author SD conceived the study.”

Reviewer #3: Definitely this is a great study and the authors hard work should be paid off. Worldwide such type of study is really limited. But the population age is a matter here. Normally within aging the gonads normally begin to do less performance. At this study, population age ranges from 40 - 70 years. Naturally, gonadal function will be decreased. But the main outcome of the study is all right, healthy individuals showed normal testosterones level than the diabetic population. If the population age is limited to 18 from 50, then more scientific evidence can be created which obviously can help on reproductive and over all life outcome. However, this study will also generate evidence. But, i found several lines and write up as out of the main flaws. Here are my comments-

INTRODUCTION:

1. Line no-61: Why loss of memory, depression, lack of concentration will come first over riding the main affects of hypogonadism? I would like to request the author to organize the line accordingly with proper citation.

2. What does the author mean by " Physical decline". Please clarify

3. Type-2 DM is written differently in all over the manuscript. I would request to use one uniform pattern.

STUDY PARTICIPANTS:

1. Cut off values for fasting blood sugar and HBA1c is adopted from where? Any reference, then author should mention it. Because different guidelines use different cut off values.

2. Study participant selection criteria should be more clear.

BODY MASS INDEX- Authors have categorized according to WHO BMI Category? But why no citation or acknowledgement?

BP MEASUREMENT- Line no 124 and whole procedure of BP measurement were done maintain a standard procedure. So, why no citation or acknowledgement?

BLOOD SAMPLE COLLECTION- This procedure is also done maintaining standard protocol. Is it? Clear writing and proper citation is very necessary here.

LABORATORY ASSAY- Are the whole cut off/ normal values are taken from standard guideline? Nothing is clear over there. If yes, then acknowledgement is required.

RESULS:

1. Most of the study participants were at the age group of 40 to 70 years. Normally gonadal function diminishes at this ages. Although a clear explanation is given on the discussion part, but i found it as not enough. It should be more clarified along with some evidences from published studies.

2. Erectile dysfunction rate is quite high and it seems normal at this age group. But the normal individual is also fall into that category. A clear explanation on this finding is very necessary to make the statement reader palatable.

DISCUSSION:

This part is quite good and have made many things clear. But still there are few scope to make it more synchronized with the findings from analysis.

Conclusion:

Few lines were beautifully written. But last portion is quite hazy and it demands re- write maintaining a flaws with the overall study.

ABSTRACT and TITLE: This is quite good and well Witten. But, again the conclusion should be more specified .

6. PLOS authors have the option to publish the peer review history of their article (what does this mean?). If published, this will include your full peer review and any attached files.

**Do you want your identity to be public for this peer review?** For information about this choice, including consent withdrawal, please see our Privacy Policy.

Reviewer #1: No

Reviewer #2: No

Reviewer #3: No

---

## [Decision Letter · Decision Letter 1]

27 Sep 2021

PGPH-D-21-00300R1

Prevalence and determinants of low testosterone levels in men with Type 2 diabetes mellitus; a case-control study in a district hospital in Ghana.

Dear Dr. Serwaa,

Thank you for submitting your manuscript to PLOS Global Public Health. After careful consideration, we feel that it has merit but does not fully meet PLOS Global Public Health’s publication criteria as it currently stands. Therefore, we invite you to submit a revised version of the manuscript that addresses the points raised during the review process.

Dear Authors, Kindly address all the points raised by the reviewers.

We look forward to receiving your revised manuscript.

Kind regards,

Palash Chandra Banik, MPhil

Academic Editor

Journal Requirements:

Additional Editor Comments (if provided):

Dear Authors

Please address the reviewers comments accordingly.

Reviewers' comments:

Reviewer's Responses to Questions

**Comments to the Author**

1. If the authors have adequately addressed your comments raised in a previous round of review and you feel that this manuscript is now acceptable for publication, you may indicate that here to bypass the “Comments to the Author” section, enter your conflict of interest statement in the “Confidential to Editor” section, and submit your "Accept" recommendation.

Reviewer #1: All comments have been addressed

Reviewer #2: All comments have been addressed

Reviewer #3: All comments have been addressed

2. Does this manuscript meet PLOS Global Public Health’s publication criteria? Is the manuscript technically sound, and do the data support the conclusions? The manuscript must describe methodologically and ethically rigorous research with conclusions that are appropriately drawn based on the data presented.

Reviewer #1: Yes

Reviewer #2: Yes

Reviewer #3: Partly

3. Has the statistical analysis been performed appropriately and rigorously?

Reviewer #1: Yes

Reviewer #2: Yes

Reviewer #3: Yes

4. Have the authors made all data underlying the findings in their manuscript fully available (please refer to the Data Availability Statement at the start of the manuscript PDF file)?

Reviewer #1: Yes

Reviewer #2: Yes

Reviewer #3: Yes

5. Is the manuscript presented in an intelligible fashion and written in standard English?

Reviewer #1: Yes

Reviewer #2: (No Response)

Reviewer #3: Yes

6. Review Comments to the Author

Reviewer #1: authors clarified and modified according to our comments

Reviewer #2: Comments to the author:

Dr. Serwaa and colleagues present a detailed case control study of 300 men with type 2 diabetes mellitus and average normal weight for low testosterone in a district hospital within a peri-urban setting, finding significant associations in this populations as shown in a prior urban setting in Ghana. Their results contribute to the literature associating type 2 diabetes and dysglycemia with male hypogonadism, in this case in a peri-urban setting of Nkenkaasu Hospital in Nkenkaasu, Ghana, West Africa in a distinct population from prior studies that contributes to the global understanding of male hypogonadism prevalence. A particular finding of interest is that both their case and control cohorts of moderate size were of normal weight on average, with their diabetes group having suboptimal glycemic control by A1c (8.06) and lower BMI (23.05) than their control group. Strengths also include measurement of pituitary hormones LH and FSH which are lower in the diabetes group, suggesting a relative secondary hypogonadism compared to the control cohort and consistent with hypogonadism occurring in type 2 diabetes. The results presented are complete, the statistical analysis is rigorous and the study was conducted following Declaration of Helsinki ethical principles. Figures 1 and 2 are particularly informative as the convey the variation in testosterone levels in relation to fasting blood glucose and HbA1c. The discussion is nicely written and cites relevant literature including articles documenting similar associations in the Ghanian population as well as the Endocrine Society guidelines as applied in this population. The authors acknowledge limitations of the study including difficulty measuring testosterone in this community due to lack of treatment access, and lack of estradiol measurement.

I appreciate the inclusion of the STROBE checklist for case-control studies, and have no additional changes to suggest to the authors.

Reviewer #3: Authors hard work and determination is praiseworthy. They have addresses all the comments raised by the reviewers and it has shaped the manuscript into a more attractive form. But still i have noticed few things are not on that standard. These comments are below

# Objectives and rationale of the study is still hazy. Last part of introduction is not well written and is giving a discrete essence.

#Sample size is less, so it is really difficult to address the title as prevalence study. It is only a proportion from small samples.

#Age group of the study may create a dilemma.

#Despite of having standard cot of values from ADA, WHO authors have used so many secondary references. I personally not in a favor to use this.

#Could not co-relate the line no 130 with the citation number 14.

#Line 193-195 demands an acknowledgement, but this is absent there i see.

#Discussion and conclusion is well in re-written form

7. PLOS authors have the option to publish the peer review history of their article (what does this mean?). If published, this will include your full peer review and any attached files.

**Do you want your identity to be public for this peer review?** For information about this choice, including consent withdrawal, please see our Privacy Policy.

Reviewer #1: No

Reviewer #2: **Yes: **Jose O. Aleman

Reviewer #3: No

---

## [Decision Letter · Decision Letter 2]

18 Oct 2021

Prevalence and determinants of low testosterone levels in men with Type 2 diabetes mellitus; a case-control study in a district hospital in Ghana.

PGPH-D-21-00300R2

Dear Dr. Serwaa,

We're pleased to inform you that your manuscript has been judged scientifically suitable for publication and will be formally accepted for publication once it meets all outstanding technical requirements.

Within one week, you'll receive an e-mail detailing the required amendments. When these have been addressed, you'll receive a formal acceptance letter and your manuscript will be scheduled for publication.

An invoice for payment will follow shortly after the formal acceptance. To ensure an efficient process, please log into Editorial Manager at https://www.editorialmanager.com/pgph/ click the 'Update My Information' link at the top of the page, and double check that your user information is up-to-date. If you have any billing related questions, please contact our Author Billing department directly at authorbilling@plos.org.

Kind regards,

Palash Chandra Banik, MPhil

Academic Editor

Additional Editor Comments (optional):

Dear Authors, Please reply the reviewer comments if any.

Reviewers' comments:

Reviewer's Responses to Questions

**Comments to the Author**

1. If the authors have adequately addressed your comments raised in a previous round of review and you feel that this manuscript is now acceptable for publication, you may indicate that here to bypass the “Comments to the Author” section, enter your conflict of interest statement in the “Confidential to Editor” section, and submit your "Accept" recommendation.

Reviewer #3: All comments have been addressed

2. Does this manuscript meet PLOS Global Public Health’s publication criteria? Is the manuscript technically sound, and do the data support the conclusions? The manuscript must describe methodologically and ethically rigorous research with conclusions that are appropriately drawn based on the data presented.

Reviewer #3: Partly

3. Has the statistical analysis been performed appropriately and rigorously?

Reviewer #3: Yes

4. Have the authors made all data underlying the findings in their manuscript fully available (please refer to the Data Availability Statement at the start of the manuscript PDF file)?

Reviewer #3: Yes

5. Is the manuscript presented in an intelligible fashion and written in standard English?

Reviewer #3: Yes

6. Review Comments to the Author

Reviewer #3: Authors hard work is praiseworthy. All comments are addressed and the new form of manuscript is better understandable.

7. PLOS authors have the option to publish the peer review history of their article (what does this mean?). If published, this will include your full peer review and any attached files.

**Do you want your identity to be public for this peer review?** For information about this choice, including consent withdrawal, please see our Privacy Policy.

Reviewer #3: **Yes: **Fardina Rahman Omi
